# POLICY OPTIMIZATION IN THE FACE OF UNCERTAINTY

## ABSTRACT

Model-based reinforcement learning has the potential to be more sample efficient than model-free approaches. However, existing model-based methods are vulnerable to model bias, which leads to poor generalization and asymptotic performance compared to model-free counterparts. In this paper, we propose a novel policy optimization framework using an uncertainty-aware objective function to handle those issues. In this framework, the agent simultaneously learns an uncertainty-aware dynamics model and optimizes the policy according to these learned models. Under this framework, the objective function can represented end-to-end as a single computational graph, which allows seamless policy gradient computation via backpropagation through the models. In addition to being theoretically sound, our approach shows promising results on challenging continuous control benchmarks with competitive asymptotic performance and sample complexity compared to state-of-the-art baselines.

## 1 INTRODUCTION

Popular reinforcement learning (RL) algorithms are divided into two main paradigms: model-free (MFRL) and model-based (MBRL) types. While achieving good asymtotic performances in many high dimensional problems (Mnih et al., 2015; Silver et al., 2017; Schulman et al., 2017; Hessel et al., 2018; Espeholt et al., 2018), MFRL methods suffer from high sample complexity since they learn state/state-action values only from rewards and do not explicitly exploit the rich information underlying the transition dynamics data. On the contrary, MBRL approaches, by trying model the transition dynamics that are in turn used for planning without having to frequently interacting with real systems, are known to have sample efficiency and thus possess more practicability (Deisenroth et al., 2013; Finn et al., 2016; Ebert et al., 2018; Sutton & Barto, 2018; Kaiser et al., 2019).

Current MBRL methods, however, still have limitations because the accuracy of the learned dynamics model is usually not satisfied, especially in complex environments (Zhang et al., 2018; Lowrey et al., 2018). The model error and its compounding effect when planning, i.e. a small bias in the model can lead to a highly erroneous value function estimate and a strongly-biased suboptimal policy, make MBRL less competitive in terms of asymptotic performance than MFRL for many non-trivial tasks. Numerous attempts have been made to tackle with this model bias problem but none of them have been really successful, such as using Gaussian Process (GP) (Deisenroth & Rasmussen, 2011; Gal & Ghahramani, 2016), Bayesian Neural Networks (Gal et al., 2016; Depeweg et al., 2016a; Kamthe & Deisenroth, 2017), and Emsembling (Kurutach et al., 2018; Clavera et al., 2018).

Another limitation of many existing MBRL methods is that they rely on the model predictive control (MPC) framework (Garcia et al., 1989). While being commonly used, MPC has serveral drawbacks (Atkeson & Schaal, 1997; Thananjeyan et al., 2019). First, each step requires solving a high-dimensional optimization problem and thus is computationally prohibitive for applications requiring either real-time or low-latency reaction such as autonomous driving. Second, the policy is only implicit via solving the mentioned optimization problem. Not being able to explicitly represent the policy makes it hard to transfer the learned policy to other tasks or to initialize agents with an existing better-than-random policy.

**Contributions.** To address those challenges of MBRL, we propose a new framework called Policy Optimization with Uncertainty-aware Model (POUM) that is able to optimize in the face of uncertainty. Our policy optimization is based on Policy Gradient, which has been widely adopted in MFRL (Lillicrap et al., 2015; Schulman et al., 2017; Haarnoja et al., 2018). However, in POUM, the objective function, a utility function, is formulated around the uncertainty-aware dynamics model. This utility function takes into account both the mean and the variance of the value function estimate. This helps reducing the model bias while effectively approximating true objective, which is the value function of the policy. For experiments, we demonstrate the advantages of POUM over state-of-the-art (SoTA) methods on various RL tasks given training from scratch and all the environments are unaltered, and also investigate on how much risk is tolerable in those tasks. And last, POUM can be represented end-to-end in a single computation graph, which greatly facilitates the training.

## 2    RELATED WORK

**Traditional MBRL.** Initial successes of MBRL in continuous control achieved promising results by learning control policies trained on models of local dynamics using linear parametric approximators (Abbeel et al., 2007; Levine & Koltun, 2013). Alternative methods such as Deisenroth & Rasmussen (2011); Levine & Koltun (2013) incorporated non-parametric probabilistic GPs to capture model uncertainty during policy planning and evaluation. While these methods enhance data efficiency in low-dimensional tasks, their applications in more challenging domains such as environments involving non-contact dynamics and high-dimensional control remain limited by the inflexibility of their temporally local structure and intractable inference time. Our approach, on the contrary, pushes the uncertainty modeling to the objective function and not anywhere else in the architecture. Plus, the fact that this objective is designed to propagate all the way to the value function makes it versatile in capturing uncertainty. What is more, all core components are constructed by neural networks gives our solution more power in dealing with high-dimensional tasks, thus acquiring asymptotically high performance compared to MFRL methods and, at the same time, retaining data efficiency in those complex domains.

**Deep Neural Networks (DNNs).** Recently, there has been a revived interest in using DNNs to learn predictive models of environments from data, drawing inspiration from ideas in the early literature on this MBRL field, mainly because the large representational capacity enables them as suitable function approximators for complex environments, especially that involve images or videos (Ebert et al., 2018; Kaiser et al., 2019). However, additional care has to be usually taken to avoid model bias, a situation where the DNNs overfit in the early stages of learning, resulting in inaccurate models. For example, Depeweg et al. (2016b) modeled a Bayesian type of DNNs to capture uncertainty in transition dynamics. In another approach, Nagabandi et al. (2017) combined a learned dynamics network with MPC to initialize the policy network to accelerate learning in model-free deep RL. Chua et al. (2018) extended this idea by introducing a bootstrapped ensemble of probabilistic DNNs to model predictive uncertainty of the learned networks and demonstrating that a pure model-based approach can attain the asymptotic performance of MFRL counterparts. However, the use of MPC to define a policy leads to poor run-time execution and hard to transfer policy across tasks. On the contrary, our framework is much simpler in that we do not employ any extra method to model the dynamics uncertainty into DNNs that are already complicated itself with numerous architectures and hyperparameters, but instead formulate a single, new uncertainty-aware objective for end-to-end optimization.

**Ensemble.** Another group of work leveraged the learned ensemble of dynamics models to train a policy network. Kurutach et al. (2018) learned a stochastic policy via trust-region policy optimization, and Clavera et al. (2018) casted the policy gradient as a meta-learning adaptation step with respect to each member of the ensemble. Buckman et al. (2018) proposed an algorithm to learn a weighted combination of roll-outs of different horizon lengths, which dynamically interpolates between model-based and model-free learning based on the uncertainty in the model predictions. To our knowledge, this is the closest work in aside from ours, which learns a reward function in addition to the dynamics function. But none of the aforementioned work propagates the uncertainty all the way to the value function and uses the concept of utility function to balance risk and return as in our model.

Finally, ensemble of DNNs also provide a straightforward technique to obtain reliable estimates of predictive uncertainty (Lakshminarayanan et al., 2017) and has been integrated with bootstrap to guide exploration in MFRL (Osband et al., 2016; Janner et al., 2019). While many of the approaches mentioned in this section employ bootstrap to train an ensemble of models, we note that their implementations comprise of reconstructing bootstrap datasets at every training iteration, which effectively trains every single data sample and thus diminishes the advantage on uncertainty quantification achieved through bootstrapping. Except for a novel objective formulation, our model is different in that, to maintain online bootstrapped datasets across ensembles, it adds each incoming data sample to a dataset according to a Poisson probability distribution (Park et al., 2007; Qin et al., 2013), thereby guaranteeing asymptotically consistent those datasets.

## 3 UNCERTAINTY-AWARE MODEL-BASED POLICY OPTIMIZATION

### 3.1 BACKGROUND

Consider a discrete-time Markov Decision Process (MDP) defined by a tuple $M = \{S, A, f, r, \gamma\}$, in which $S$ is a state space, $A$ is an action space, $f : S \times A \to S$ is a deterministic (or probabilistic) transition function, $r : S \times A \to \mathbb{R}$ is a deterministic reward function, and $\gamma \in (0, 1)$ is a discount factor. We define the return as sum of the rewards $r(s_t, a_t) = r(s_t, \pi(s_t))$ for $t = 0, \ldots, T$ for the whole trajectory $(s_0, a_0, ..., s_T, a_T)$ induced by a policy $\pi : S \to A$ and discounted by $\gamma$. Here $T \in \mathbb{Z}_+$ is a task horizon, which may take a value of $\infty$ for non-episodic environments. The goal of RL is to find an optimal policy $\pi^\star$ to maximize the expected return

$$J(\pi) = \mathop{\mathbb{E}}_{s_0 \sim S} [V^\pi(s_0)] \tag{1}$$

where the value function is defined as $V^\pi(s_0) = \sum_{t=0}^{T-1} \gamma^t r(s_t, \pi(s_t))$, and the state transition is $s_{t+1} = f(s_t, \pi(s_t))$, with $s_0$ being randomly chosen from the distribution of $s \in S$. Then if the dynamics function $f$ and the reward function $r$ are given, solving Equation 1 can be done using the Calculus of Variations (Young, 2000) or using Policy Gradient (Sutton et al., 2000) when the control function is parameterized or is finite dimensional.

In RL, however, $f$ and $r$ are often unknown and hence Equation 1 becomes a blackbox optimization problem with an unknown objective function. Following the Bayesian approach commonly used in the blackbox optimization literature (Shahriari et al., 2015), we propose to solve this problem by iteratively learning a probabilistic estimate $\widehat{V}$ of $V$ from data and optimizing the policy according to this approximate model, as detailed in the next section.

### 3.2 FORMULATION OF UNCERTAINTY-AWARE OPTIMIZATION OBJECTIVE

It is worth noting that any unbiased method would model $\widehat{V}(\pi)$ as a probabilistic estimate, i.e. $\widehat{V}(\pi)$ would be a distribution (as opposed to a point estimate) for a given policy $\pi$. Optimizing a stochastic objective is, however, not well-defined. Our solution is to transform $\widehat{V}$ into a deterministic utility function that reflects a subjective measure balancing the risk and return. Following Markowitz (1952); Sato et al. (2001); García & Fernández (2015), we propose a risk-sensitive objective criterion using a linear combination of the mean and the standard deviation of $\widehat{V}(\pi)$. Formally stated, our objective criterion, which we also call the utility function, now becomes

$$U(\pi)(s_0) = \mathop{\mathbb{E}}_{s_0 \sim S} \left[ \mu\left(\widehat{V}(\pi)(s_0)\right) + c \times \sigma\left(\widehat{V}(\pi)(s_0)\right) \right], \tag{2}$$

where $\mu$ and $\sigma$ are respectively the mean and the standard deviation of $\widehat{V}(\pi)(s_0)$, and $c$ is a constant that represents the subjective risk preference of the learning agent. A positive risk preference infers that the agent is adventurous while a negative risk preference indicates that the agent has a safe exploration strategy. To our best knowledge, this uncertainty-aware model-based objective function has not been used in the RL literature.

### 3.3 EMPIRICAL ESTIMATE OF VALUE FUNCTION

Section 3.2 provides a general framework for policy optimization under uncertainty, assuming the availability of the estimation model $\widehat{V}(\pi)$ of the true value function $V(\pi)$. In this section, we

describe how to estimate $\widehat{V}(\pi)$ with a model-based approach. The main idea is to approximate the functions $\{f, r\}$ with probabilistic parametric models $\{\widehat{f}, \widehat{r}\}$ and fully propagate the estimated uncertainty when planning under each policy $\pi$ from an initial state $s_0$. The value function estimate $\widehat{V}$ can be formulated as

$$\widehat{V}(\pi)(s_0) = \sum_{t=0}^{T-1} \gamma^t \widehat{r}(\hat{s}_t, \pi(\hat{s}_t)), \tag{3}$$

where $\hat{s}_0 = s_0$ and $\hat{s}_{t+1} = \widehat{f}(\hat{s}_t, \pi(\hat{s}_t))$ for $t = 0, \ldots, T-1$. Next, we describe how to efficiently model $\{f, r\}$ with well-calibrated uncertainty and a rollout technique that allows the uncertainty to be faithfully propagated into $\widehat{V}(\pi)$.

### 3.3.1 Bootstrap Setup for Model Learning

Following the traditional bootstrap methodology, the empirical model function $\widehat{f}$ is represented as $\{\widehat{f}_{\phi_k}(s_t, a_t) \rightarrow s_{t+1}\}_{k=1}^B$. For simplicity of implementation, we model each bootstrap replica as deterministic and rely on the ensemble as the sole mechanism for quantifying and propagating uncertainty. Each bootstrapped model $\widehat{f}_{\phi_k}$, which is parameterized by $\phi_k$, learns to minimize the $L2$ one-step prediction loss over the respective bootstrapped dataset $\mathbb{D}_k$:

$$\min_{\phi_{k:=1 \mapsto B}} \mathbb{E}_{(s_t, a_t, s_{t+1}) \sim \mathbb{D}_k} \|s_{t+1} - \widehat{f}_{\phi_k}(s_t, a_t)\|_2^2. \tag{4}$$

The training dataset $\mathbf{D}$, from which the bootstrapped datasets $\{\mathbb{D}_k\}_{k=1}^B$ are sampled, stores the transitions on which the agent has experienced. Since each model observes its own subset of the real data samples, the predictions across the ensemble remain sufficiently diverse in the early stages of the learning and will then converge to their true values as the error of the individual networks decreases.

In addition to model estimation and unlike many other model-based approaches, we also learn the reward function along the same design of classical MBRL algorithms Sutton (1991). But in POUM, we use a deterministic model (also parameterized by a DNN) for the reward function to simplify the policy evaluation.

### 3.3.2 Bootstrap Rollout

In this section, we describe how to propagate the estimates with uncertainty from the dynamics model to evaluate a policy $\pi$. We represent our policy $\pi_\theta : S \rightarrow A$ as a neural network parameterized by $\theta$ . Note that we choose to represent our policy as *deterministic*. We argue that while all estimation models, including that of the dynamics and of the value function, need to be stochastic (i.e. uncertainty-aware), the policy does not need to be. The policy is not an estimator and deterministic policy simply means that the agent is consistent when taking an action, no matter how uncertain it may know about the world.

Given a deterministic policy $\pi_\theta$ and an initial state $s_0 \in \mathbb{D}$, we can estimate the distribution of $V(\pi)(s_0)$ by simulating $\pi_\theta$ through each each bootstrapped dynamics model. And since each bootstrap model is an independent approximator of the dynamics function, by expanding the value function via these dynamics approximators, we eventually obtain independent estimates of that value function. Finally, those separate and independent trajectories collectively form an ensemble estimator of $V$.

In practice, we sample these trajectories with a finite horizon $H < T$. It is still a challenge to expand the value function estimation for a very long horizon due to a few reasons. First, DNNs training becomes harder when the depth increases. Second, despite our best effort to control the uncertainty, we still do not have a guarantee that our uncertainty modeling is perfectly calibrated, which in turn may be problematic if the planning horizon is too large. Finally, policy learning time is proportional to the rollout horizon.

### 3.4 POLICY GRADIENT

Based on Equation 2, the optimization target to optimize based on policy gradient method is:

$$\arg\max_{\theta} J(\theta) = \mathbb{E}_{s \sim S} \left[ U_\theta(s) \right], \tag{5}$$

where $U_\theta(s) = \mu(\widehat{V}_\theta(s)) + c \times \sigma(\widehat{V}_\theta(s))$. Using the ensemble method and the rollout technique described above, we can naturally compute $\mu(\widehat{V}_\theta(s))$ and $\sigma(\widehat{V}_\theta(s))$ for a given policy $\pi_\theta$ and for a given state $s$. Therefore, the policy $\pi_\theta$ can be updated using the SGD or a variance of it.

Importantly, in terms of implementation, it is worth noting that the aforementioned rollout method also allows for easily expressing $U(\theta)$ in Equation 5 as a single computational graph of $\theta$. This makes it straightforward to compute the policy gradient $\nabla_\theta U_\theta(s)$ using automatic differentiation, a feature provided by default in most popular deep learning toolkits.

## 4 ALGORITHM SUMMARY

---

**Algorithm 1** Policy Optimization with Uncertainty-aware Model (POUM)

---

1: Initialize a training dataset $\mathbf{D}$, bootstrapped datasets $\{\mathbb{D}_i\}_{i=1}^{B}$, parameterized bootstrapped models $\{\widehat{f}_i\}_{i=1}^{B}$, a parameterized reward model $\hat{r}_\phi$, and a parameterized deterministic policy $\pi_\theta$.
2: **while** not done **do**
3:   • Step in the environment, collect new data point $(s, a, s', r)$ and push into $\mathbf{D}$,
4:   • Sample from $\mathbf{D}$ and push data into the bootstrapped replay buffers: for each member $i^{th}$ in the ensemble, add $z_i \sim Poisson(1)$ copies of that data point to $\mathbb{D}_i$,
5:   • Update $\{\widehat{f}_i\}_{i=1}^{B}$ on $\mathbb{D}_i$ and $\hat{r}_\phi$ on $\mathbf{D}$ using SGD,
6:   • Evaluate $\widehat{V}_\theta(s)$ and $U_\theta(s)$ by simulating through the learned models $\{\widehat{f}_i\}_{i=1}^{B}$ and $\widehat{r}_\phi$,
7:   • Update $\pi_\theta$ using SGD with the policy gradient being backpropagated on $\mathbb{E}_s \left[ U_\theta(s) \right]$ through the learned models.
8: **end while**

---

We summarize our framework POUM in Algorithm 1 and later in this section, we will also highlight some important details in our implementation.

### 4.1 DYNAMICS MODEL LEARNING WITH ONLINE BOOTSTRAP

As discussed in Section 1, there are several prior attempts to learn uncertainty-aware dynamics models such as GPs, Bayesian neural networks (NNs), dropout NNs and ensemble of NNs. In this work, however, we employ an ensemble of bootstrapped DNNs. Bootstrap is a generic, principled and statistical approach for uncertainty quantification. Furthermore, as will be also later explained in Section 3.4, this ensembling approach also gives rise to easy gradient computation.

#### 4.1.1 ONLINE BOOTSTRAP FOR TRAINING DATA

Bootstrap learning is often studied in the context of batch learning. However, since our agent updates its empirical model $\widehat{F}$ after each physical step for the best possible sample efficiency, we follow an online bootstrapping method by sampling from Poisson distribution (Oza, 2005; Qin et al., 2013). This is a very effective online approximation to batch bootstrapping, and can be easily done by this simple rule: bootstrapping a dataset $\mathbb{D}$ with $n$ examples means sampling $n$ examples from $\mathbb{D}$ with replacement. In detail, each example $i$ will appear $z_i$ times in the bootstrapped sample where $z_i$ is a random variable whose distribution is $Binom(n, 1/n)$ because during resampling, the $i^{th}$ example will have $n$ chances to be picked, each with probability $1/n$. This $Binom(n, 1/n)$ distribution converges to $Poisson(1)$ when $n \to \infty$. Therefore, for each new data point, this method adds $z_k$ copies of that data point to the bootstrapped dataset $\mathbb{D}_k$, where $z_k$ is sampled from a $Poisson(1)$.

**Online off-policy learning.**   Except for the initialization step (we may initialize the models with batch training from off-policy data), our model learning is an online learning process. For each time

step, the learning cost stays constant and does not grow over time, which is required for lifelong learning. Despite being online, the learning is off-policy because we maintain a bootstrapped replay buffer for each model in the ensemble. For each model update, we sample a minibatch of training data from the respective replay buffer. In addition, as mentioned, the models can also be initialized from existing data even before the policy optimization starts.

### 4.1.2 LINEARLY-WEIGHTED SAMPLING FROM BOOTSTRAPPED TRAINING DATA

Since our replay buffers are accumulated online, a naive uniformly sampling strategy would lead to early data being sampled more frequently than the later ones. We thus propose a linearly weighted random sampling scheme to mitigate this early-data bias issue. In this sampling scheme, example $i^{th}$ is randomly sampled with weight $i$, i.e. higher weights for the fresher examples in each online update step. Despite its simplicity, this scheme plays an important role in data bias removal, as shown in Appendix A.1.

## 5 EXPERIMENT

Our experiments are designed to help 1) compare our POUM framework with other SoTA approaches and 2) investigate the impact of the risk factor in our utility function on guiding agents.

### 5.1 COMPARISON TO BASELINE ALGORITHMS

**Experimental Design.** We evaluate the performance of our POUM algorithm on four continuous control tasks including: one classic control task (Pendulum-v0) and three other tasks in the MuJoCo simulator (Todorov et al., 2012) from OpenAI Gym (Brockman et al., 2016). It is important to note that, we keep the default configurations prodived by OpenAI Gym (See Appendix A.2.1) and also does not assume access to the reward function as some recent works in model-based reinforcement learning (Chua et al., 2018; Clavera et al., 2018; Kurutach et al., 2018).

For the baselines, we compare POUM to the following SoTA algorithms designed for continuous control: MBPO (Janner et al., 2019), DDPG (Lillicrap et al., 2015), SAC (Haarnoja et al., 2018), STEVE (Buckman et al., 2018). For each one of them, we evaluate the learned policy after every episode. The evaluation is done by running the current policy on 20 random episodes and then computing the average return over them.

**Results.** Figure 5.1 shows that POUM has a sample efficiency compared to the baseline algorithms across a wide range of environments. Furthermore, it also has the asymptotic performance competitive to or even better than that of the model-free counterparts. Note that, there are horizontal parts at the beginning of evaluation curves in some algorithms and environments, that because these algorithms take random exploration at the beginning of training (as their default configuration) to initialize dynamics. For simple environments: Pendulum-v0, Reacher-v2, Push-v2, our POUM can get a good performance without initialized dynamics[1].

However, Figure 5.1 also shows that the performance of POUM in more complex environment like HalfCheetah-v2 is sensitive to random seeds. We hypothesize that this is due to the impact of risk-preference value on policy optimization framework and our strategy of aggressive online learning and linearly weighted batch sampling. The ablation study below validates our current analysis on these hypotheses.

### 5.2 ABLATION STUDY

To obtain a better understanding about the role of the subjective risk preference in the utility function, we conduct an ablation study on the parameter $c$ that controls this risk factor (Equation 2) and make the following observations.

As illustrated in Figure 5.2, as the first observation, complete zero risk is not a good choice. This behavior is expected because no risk means no uncertainty is quantified properly, leading to agents

---

[1]MBPO failed to attain a good performance for Reacher-v2 and Pusher-v2, regardless our best effort to produce the results based on the authors' official repository.

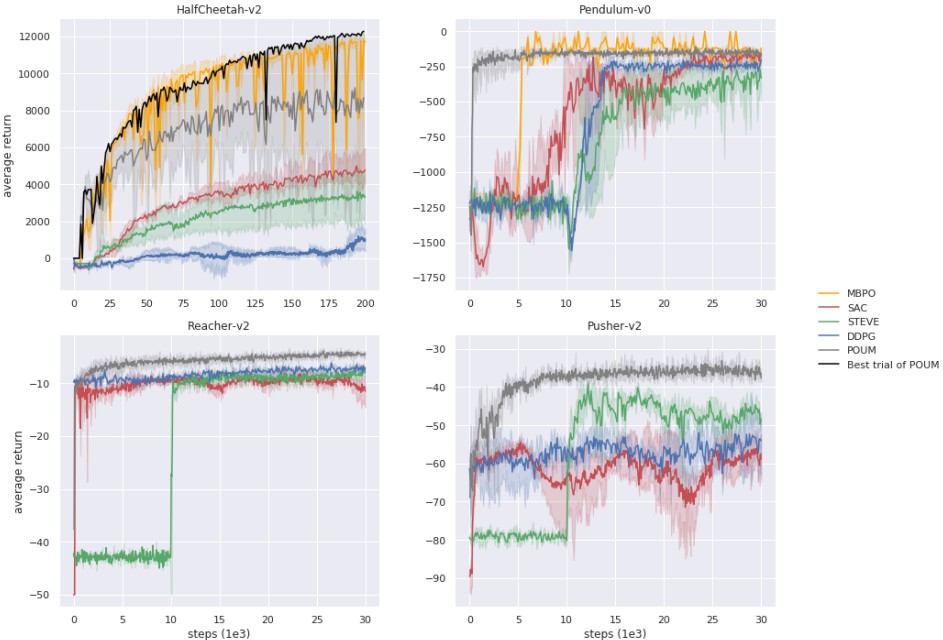

Figure 1: Average return of POUM model over 3 different randomly selected random seeds compared with SoTA appoaches. Solid lines indicate the mean and shaded areas indicate one standard deviation. POUM beats all other solutions on environments tested, except for HalfCheetah-v2 where it has a competitive performances compared to MBPO.

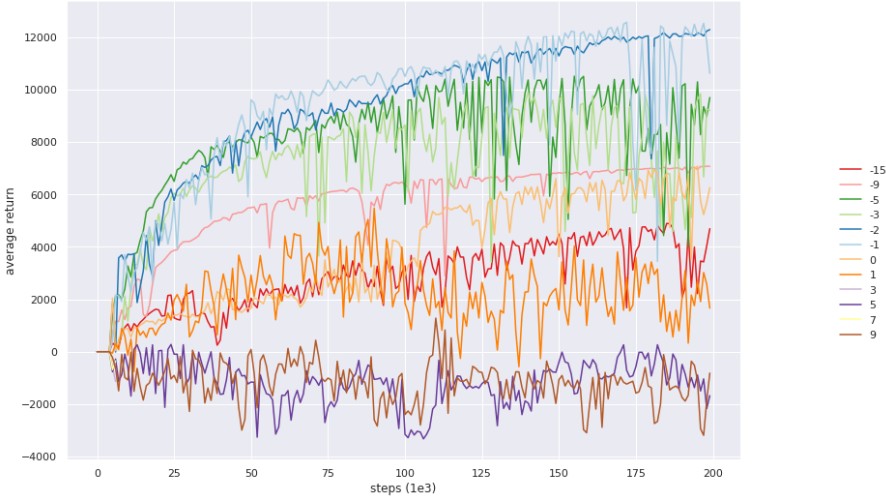

Figure 2: POUM with different subjective risk preference values on HalfCheetah-v2 environment. The moderate risk yields the best return while too much risk (either too high or too low) will harm the agent in getting good results. We use the same settings of HalfCheetah-v2 environment as presented in Appendix A.2.3 excluding risk-preference values

could not learn well to model the dynamics, as well as optimize an efficient policy. In another observation, POUM performs best with the risks $c = -1$, follows by $c = -2$ and $c = -3$, while it gets worse and worse at both directions, the risk factor goes either higher or lower. This phenomenon is because the risk factor controls the scale of standard deviation, and hence the variance. Too low or too high risks, consequently, imply too much variance which is not favorable in many cases because

it indudes agents to explore more aggressively and hence suffer more potential failures, while not exploiting current, safer experiences. Finally and interestingly, as the scale of the risk changes, both directions are not behaving the same. In particular, POUM gets bad results with positive risk-preference value, and even can not learn with high positive value. That is because in current work, we use fixed the subjective risk preference value and at the beginning of learning process, the dynamics models are unstable and high variance, with high positive risk-preference values, policy learning strange decisions. In contrast, with negative risk-preference, our utility function work as lower confidence bound that keep policy in a safe region. The figure indicates that with lower negative risk-preference value, the learning curve is more stable. However, lower risk-preference value means that less exploration, and results in lower final reward.

## 6 DISCUSSION AND CONCLUSION

In summary, this paper proposed a new approach in MBRL in which we developed a novel objective function that balances the mean and variance in the estimation of the value function, which is induced by the model. Our experiments suggest that our POUM algorithm not only can achieve the asymptotic performance of model-free methods in challenging continuous control tasks and compared to other SoTA approaches, it does so in much fewer samples. We further demonstrate that the model bias issue in model-based RL can be dealt with effectively with principled and careful uncertainty quantification, by guiding agents with a subjective risk factor. Unlike other methods, quantifying and controlling the uncertainty with a novel uncertainty-aware objective function, and without any complex designs for other components is an advantage, of being simple yet efficient, compared with others.

Nonetheless, we acknowledge that our current implementation for POUM still has several limitations, such as high variance in the empirical performance, which still depends on many hyper-parameters (plan horizon, risk sensitivity, and all hyper-parameters associated with neural networks training techniques) and even depends on random seeds. It is, however, worth noting that these traits are not unique to our methods. In spite of this limitation, the results indicate that if implemented properly, MBRL methods can be both sample efficient and have better asymptotic performances than the MFRL counterparts on challenging tasks. In addition, by explicitly representing both the dynamics model and the policy, POUM enables transfer learning, not just for the world (dynamics) model but also for the policy.

To sum up, we identify that sample efficiency, off-policy learning, and transferability are three necessary, albeit not sufficient, properties for real-world reinforcement learning. We claim that our method meets these criteria and hence is a step towards real-world reinforcement learning.

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

# A    APPENDIX

## A.1    WHY LINEARLY WEIGHTED RANDOM SAMPLING IS A FAIRER SAMPLING SCHEME

Consider the following online learning process: for each time step, we need to randomly sample an example from the accumulating dataset. Suppose that at time $t$, each example $i^{th}$ is randomly sampled with weight $w(t, i)$. Note that at each time $t$, we have a total of $t$ examples in the dataset. Then the probability of that example being sampled is

$$\frac{w(t, i)}{\sum_{k=1}^{t} w(t, k)}.$$

If we use uniformly random sampling then the expected number of times an example $i^{th}$ gets selected until time $t$ is

$$C_i^t = \sum_{k=i}^{t} \frac{1}{k}.$$

Hence, for all $t$, for $i > j$, $C_i^t$ is larger than $C_j^t$ by $\sum_{k=i}^{j} \frac{1}{k}$. Now, if we use a linearly weighted random sampling scheme, in which $w(t, i) = i$, then the expected number of times an example $i^{th}$ gets selected until time $t$ is

$$C_i^t = \sum_{k=i}^{t} \frac{2i}{k(k+1)} = 2 \sum_{k=i}^{t} \left( \frac{i}{k} - \frac{i}{k+1} \right) = 2 - 2\frac{i}{t}.$$

We can see that at time $t$, $C_i^t$ is still larger than $C_j^t$ for $i < j$ but by weighting recent examples more in each online update step, we reduce the overall early-data bias.

## A.2    EXPERIMENTAL SETTINGS

### A.2.1    ENVIRONMENTS

Table 1: Description of the environment used for testing

| Environment | State dimension | Action dimension | Task horizon |
|---|---|---|---|
| Reacher-v2 | 11 | 2 | 50 |
| Pusher-v2 | 23 | 27 | 100 |
| Pendulum-v0 | 3 | 1 | 200 |
| HalfCheetah-v2 | 23 | 6 | 1000 |

### A.2.2    NETWORK ARCHITECTURE

### A.2.3    HYPER-PARAMETER SETTINGS

Table 2: Network layer configurations

| Environment | bootstrap model: $\widehat{f}_i$ | Reward model: $\hat{r}_\phi$ | Policy: $\pi_\theta$ |
|---|---|---|---|
| Reacher-v2 | $[13, 256, 128, 11]$ | $[13, 256, 128, 1]$ | $[11, 128, 128, 2]$ |
| Pusher-v2 | $[30, 1024, 512, 256, 23]$ | $[30, 1024, 512, 256, 1]$ | $[23, 512, 256, 7]$ |
| Pendulum-v0 | $[4, 64, 64, 3]$ | $[4, 64, 64, 1]$ | $[3, 8, 1]$ |
| HalfCheetah-v2 | $[23, 1024, 512, 256, 17]$ | $[23, 1024, 512, 256, 1]$ | $[17, 512, 256, 6]$ |

Network format: [ input size, hidden layers size, output size ]

Table 3: Hyper-parameter settings

| Hype-parameter | Pendulum-v0 | Reacher-v2 | Pusher-v2 | HalfCheetah-v2 |
|---|---|---|---|---|
| Bootstrap size | 16 | 8 | 16 | 16 |
| Plan horizon | 20 | 30 | 35 | 20 |
| Risk preference | -0.25 | -0.25 | 0 | -1 |
| Policy updates (per environment step) | 8 | 4 | 16 | 2 |
| Dynamics update (per environment step) | 8 | 6 | 6 | 6 |

