# OpenReview forum: "Policy Optimization In the Face of Uncertainty"
_ICLR.cc/2020/Conference — Reject_

### Official Review · AnonReviewer2 · 2019-10-17
**Official Blind Review #2**

**Rating:** 3

**Review:**

I enjoyed this paper overall, and I think the idea is a good one. However there remain significant issues with the paper that preclude me giving a good score. Firstly, there is almost no discussion of the environment model. Anyone who has worked on Model Based RL will tell you that the details here are crucial. This deserves a full discussion, and a comparison to other methods in the literature.

Next, the experimental results really aren't convincing. The dependence on random seeds is worrying, and isn't as common in model free algorithms as you claim, which are mostly robust to seeds (the good ones at least). The fact that the best policy is risk *averse* is very strange, since these are estimates combining both the epistemic and aleatoric uncertainty (which is somewhat unfortunate), which means being risk averse would lead the agent to not explore. That is very worrying and makes me think that something very strange is going on with the models. In fact since the policy is deterministic and the environment / rewards are practically speaking deterministic, the uncertainty here is actually mostly epistemic and so a c < 0 means the agent is disincentivized from exploring.

There should be more discussion about the fact that the policy is deterministic. Is this merely to make estimating V^pi easier?

"Next, we provide a convergence convergence analysis and show that maximizing this utility function
U(π) is equivalent to maximizing the unknown value function V (π)."

Word convergence appears twice in a row, but more importantly this is totally missing! Where is the analysis?

In the algorithm you write:
"Update {fb} and rˆφ using SGD"
But on what data? Presumably sampled from D but this isn't mentioned.

Is it the case that the policy is updated *only* using the model based rollouts? I.e., the reward signal is never used in the policy gradient but only used to train the models? If so, this seems quite fragile and I would like to see a comparison of different approaches here.

Table 2 is unreadable and needs to be explained.

It would appear that you are missing a reference to the very relevant UBE paper, which also deals explicitly with the uncertainty of the value function estimates: https://arxiv.org/abs/1709.05380
In fact I would be curious to see any way that these two approaches could be combined (though that would be follow up work).


**Experience Assessment:**

I have published in this field for several years.

**Review Assessment: Checking Correctness Of Derivations And Theory:**

N/A

**Review Assessment: Checking Correctness Of Experiments:**

I assessed the sensibility of the experiments.

**Review Assessment: Thoroughness In Paper Reading:**

I read the paper at least twice and used my best judgement in assessing the paper.

---

> ### Author Response · Authors · 2019-11-11
> **Response to review**
>
> Thank you for taking time to review our paper, for your comment.
>
> ==========================
> # Model Details
> ==========================
> Re: model details. It is indeed that the model details are important in practice. We chose to not highlight the model details to emphasize what we think more novel and more important: the policy optimization framework based on an uncertainty-aware model.
>
> For our model, it is not very different from [PETS, MBPO] except that:
> - We use bootstrap learning instead of ensemble learning: we don’t just learn models with different initializations but those models are also learned from different datasets.  In our work, we only maintained and grew different datasets corresponding to different models completely independently.
> - We don't model the aleatoric uncertainty. In fact, in the MuJoCo environments that we experiment, the true dynamics is fully deterministic and there's no aleatoric uncertainty.
>
> ==========================
> #Dependence on random seeds:
> ==========================
> It also worries us that the variance of our method is quite high. It's the top priority that we would like to address in future work. So in this paper, we’d rather highlight the fact that our novel and principled method can yield great results, although some implementation details still need to be polished.
>
> ===========================
> #The best policy is risk *averse*:
> ===========================
> The risk-preference “c” being negative doesn't quite mean that the agent is not exploring. It just means that the agent still explores but is more conservative when exploring.
>
> Said differently, our best agents, in our limited experimental studies, are the ones that do safe exploration.
>
> ==================
> #Deterministic policy:
> ==================
> Making V(pi) easier was not the main point for treating \pi deterministic.
>
> We simply think that there's no good reason for treating \pi stochastic. \pi being deterministic simply means that the decision making, for each situation, is consistent. A rational agent should not make a decision that is dependent on coin tossing. Even a highly exploratory strategy should be deterministic and purely conditional on the current state.
>
> ===============
> #Other questions:
> ===============
> In the algorithm, we update bootstrap model f^{b}_i on bootstrap dataset D_i, and update r_{\phi} on the single replay buffer D.
>
> The reward data is only used to learn the reward model, which in turn is used in the value function estimate (the policy update relies on computing the gradient of the V_{\pi}(s) as a function of \pi). In estimating V_{\pi}(s), we generate trajectories using the dynamics model. However, the estimated rewards along those trajectories are needed to compute the value function.
>
> Explanation of Table 2:
> f_i means each bootstrap network for learning the dynamics model
> r_{\phi} means reward network.
> Policy \pi_{\theta}.
> Example for environment Reacher-v2: [13, 256, 128, 11] means:
> Input of network has size of 13.
> Output of network has size of 11.
> Network contains two hidden layers with size of 256, 128 consequently.
>
> =========
> #Summary:
> =========
> To the best of our literature research, this is the first MBRL work that formulates the objective, fully represented by the model and the policy, as a utility function (that balances return and risk).  The optimization is consistent with that objective function and the gradient is computed can be computed analytically (via backprop through the models).

---

### Official Review · AnonReviewer3 · 2019-10-20
**Official Blind Review #3**

**Rating:** 3

**Review:**

Summary:
The main contribution of this work is introducing the uncertainty-aware value function prediction into model-based RL, which can be used to balance the risk and return empirically.

Methodology
This work uses a linear combination of the mean and standard deviation of value function to capture the uncertainty in learning state value function.

It is not clear how to convert the objective function from Eq 2 (expectation over the initial state) to Eq 5 (expectation over all states). Those two objectives are not equal.

It is not clear how does the uncertainty in model prediction (dynamics and reward function)
can be alleviated through the proposed method, as claimed in the introduction.
It seems the novelty part lies in considering the uncertainty of value function estimation.
How does this relate to solving the limitation of model predictive control?

What is the objective function for learning reward function r_\phi?


Experimental results:
The experiments are not sufficient to demonstrate the effectiveness of the proposed method.
It would be more convincing to compare the proposed method with a few more model-based approaches on more tasks. The results of MBPO is better
the proposed POUM in one of two tasks. The performance of
MBPO on Reacher-v2 and Pusher-v2 is missing?


Writing:
This paper has many typos and the presentation is not very clear.
- Section 4.1 "in Section 3.4,"
- Last paragraph in P3: convergence convergence analysis
- "shows that POUM has a sample efficiency compared"


**Experience Assessment:**

I do not know much about this area.

**Review Assessment: Checking Correctness Of Derivations And Theory:**

N/A

**Review Assessment: Checking Correctness Of Experiments:**

I assessed the sensibility of the experiments.

**Review Assessment: Thoroughness In Paper Reading:**

I made a quick assessment of this paper.

---

> ### Author Response · Authors · 2019-11-11
> **Response to review**
>
> Thank you for taking time to review our paper, and for your feedback!
>
> ========================================================================
> #How is the uncertainty in models alleviated through the proposed method:
> ========================================================================
> As we describe in Section3.3.2: Given policy \pi_{\theta} and an initial state s_0, We can estimate the distribution of V(\pi)(s) by simulating \pi through each bootstrapped dynamic model. Since each bootstrap model as an independent approximator of the dynamics function, by expanding the value function via these dynamics approximators, we obtain independent estimates of the value function. These separate and independent trajectories collect tively form an ensemble estimator of V(\pi)(s).
> Consequently,  the uncertainty of models is propagated into the estimate of V(\pi)(s).
>
> # How we estimate V(\pi)(s) has no connection to MPC.
> Our overall method resolves the limitations of MPC by representing the optimal policy by a parametric policy (neural network). We optimize this policy using policy gradient computation. Unlike other traditional policy gradient methods, which are mostly model-free, our method computes the policy gradient by estimating V(\pi) analytically and then use backpropagation for computing the gradient. Our method allows computing the gradient for an arbitrary point s in S since V(\pi) is fully expressed via the models (dynamics model, policy model, and reward model).
>
> =======================
> # Equation 2 and Equation 5:
> =======================
> We are sorry about the confusion. The two are in fact equivalent. In both equations, the objective function is computed by taking the expectation of the value function (or utility function) evaluated at s_0, which is sampled from the entire state distribution.
>
> ===============
> #Reward function:
> ===============
> The objective function of reward function R_\phi:
> min E(s, a ,r)~D||r - R_\phi(s, a)||2
> which the same with objective function for learn each bootstrap f_i in dynamics model. The different thing is that bootstrap function f_i is trained on bootstrap data D_i while R_\phi is trained on main replay_buffer D.
>
> ===================
> #Experimental results:
> ==================
> We compare our algorithm to two model-based methods: MBPO and STEVE. To our knowledge, MBPO is  currently the algorithm that provides the best experimental results on standard OpenAI Gym mujoco environments.
> There are also other works on MBRL including ME-TRPO [Kurutach et al., 2018], MB-MPO [Clavera et al., 2018], PETS [Chua et al., 2018] ..., but their public implementations couldn’t run on standard OpenAI gym environments that we rely on. For instance, several of them relied on the modified MuJoCo environments in RLLab, which makes it easier to train an agent on Half Cheetah.
>
> Hence, we couldn’t do a fair comparison against those methods.
>
> As an aside, this paper [https://arxiv.org/abs/1907.02057v1] is another example of why benchmarking many model-based RL methods is not easy and has been published as a standalone paper. Even in that work, the comparisons are not entirely fair. This GitHub issue shows an example: [https://github.com/WilsonWangTHU/mbbl/issues/2]
>
> We did not include the results of MBPO on Pusher and Reacher because we couldn’t find good hyper-parameters for MBPO on these environments. Our best-effort hyperparameters for MBPO didn’t yield good results at all.
>
> Although the variance of our method is quite high, in this paper, we’d rather highlight the fact that our novel and principled method can yield great results.
>
> =========
> #summary:
> =========
> To the best of our literature research, this is the first MBRL work that formulates the objective, fully represented by the model and the policy, as a utility function (that balances return and risk).  The optimization is consistent with that objective function and the gradient is computed can be computed analytically (via backprop through the models).

---

### Official Review · AnonReviewer1 · 2019-10-24
**Official Blind Review #1**

**Rating:** 1

**Review:**

# Introduction
The motivation of this paper relies on the dynamics model not being accurate enough, which leads to compounding errors. Hence, a proper characterization of the uncertainty is needed. However, the model-based methods that suffer the most of this problem are the ones that are build on top of policy gradients (since they need to predict the entire trajectory) (e.g., [6]). The methods that learn a Q-value function from the model do not suffer as much from this problem since they just predict shorter horizons. Current model-based RL methods that learn a Q or value-function take into account the uncertainty (i.e., STEVE, MBPO). Those methods are not “less competitive in terms of asymptotic performance.”

There has been work on learning a parametric policy from MPC. Therefore, you can extract a parametric policy from the optimization that MPC performs. The statement “Not being able to explicitly represent the policy makes it hard to transfer the learned policy to other tasks or to initialize agents with an existing better-than-random policy” is not true. The MPC will transfer better to other tasks that have the same dynamics, since it is not task specific. Given the learned dynamics model and the reward function you can act optimally in any new task as long as the learned dynamics are valid.

# Related work
My main concern with the related work section is that there a lot of literature on risk sensitive and optimism in the face of uncertainty (which is a subset of your method when c>0) in control, bandits, and some on *reinforcement learning* that has been neglected.

# Uncertainty-Aware Model-Based Policy Optimization
As said before, risk-sensitive in reinforcement learning has been done before and there’s even more work on control and bandits. For instance in [1] (page 5, paragraph (b)) has the same equation and they discuss the effect of the constant being negative or positive. More recent work has also used similar formulations [2].
This section mostly contains previous work, e.g., bootstrap rollout, policy gradient, using a deterministic policy ([3, 4, 5]). One thing that it’s still not clear from reading the paper, are you backpropagating the through the dynamics model, are you using a policy gradient method (REINFORCE, TRPO, PPO,… )?

# Algorithm Summary
One of the novelties introduced is the fact that the data used in each model comes from sampling a Poisson variable. However, this is not ablated in the results sections. Is it necessary? [6] Claims that there’s no need to use different data for the learned models.

# Experiment
The experiment section lacks from more complex environments, in this case the most complex is half-cheetah. Furthermore, given that 3 of the tasks are short horizon tasks you should probably also compare against model-based methods that build on top of policy gradients (e.g., [6]).

It seems that some choices in the algorithm are not ablated: 1) use of poisson, 2) use of deterministic vs stochastic policy, 3) Is there a single risk that works across environments? Which environments are risk prone/adverse? 4) How about having c ~ N(0, 1), effectively modelling V as a gaussian?

-----------------------------------

Overall, the paper is not mature enough to be accepted: there is not enough novelty, and the results lack of novelty, enough delta in performance from prior work, and have high variance.

------------------------------------

Minor/Typos:
First paragraph: “trying model the transition”
What does it mean that the accuracy is not satisfied?
Why the related work on deep model-based reinforcement learning is called Deep Neural Networks?
3.2 third paragraph: “Next we provide a convergence convergence …”
3.3.2, first paragraph: “no matter how uncertain it may know about the world”
Why the axis in the results section mean different things?



[1] Risk-sensitive Reinforcement Learning. Yun Shen, Michael J. Tobia, Tobias Sommer, Klaus Obermayer.
[2] Plan Online, Learn Offline: Efficient Learning and Exploration via Model-Based Control. Kendall Lowrey, Aravind Rajeswaran, Sham Kakade, Emanuel Todorov, Igor Mordatch
[3] Continuous control with deep reinforcement learning. Timothy P. Lillicrap et. al.
[4] Model-Based Value Estimation for Efficient Model-Free Reinforcement Learning. Vladimir Feinberg, Alvin Wan, Ion Stoica, Michael I. Jordan, Joseph E. Gonzalez, Sergey Levine
[5] Sample-Efficient Reinforcement Learning with Stochastic Ensemble Value Expansion. Jacob Buckman, Danijar Hafner, George Tucker, Eugene Brevdo, Honglak Lee.
[6] Model-Ensemble Trust-Region Policy Optimization. Thanard Kurutach, Ignasi Clavera, Yan Duan, Aviv Tamar, Pieter Abbeel.

**Experience Assessment:**

I have published in this field for several years.

**Review Assessment: Checking Correctness Of Derivations And Theory:**

I did not assess the derivations or theory.

**Review Assessment: Checking Correctness Of Experiments:**

I carefully checked the experiments.

**Review Assessment: Thoroughness In Paper Reading:**

I read the paper thoroughly.

---

> ### Author Response · Authors · 2019-11-12
> **Response to review**
>
> Thank you for taking time to review our paper and for the feedback.
>
> First of all, we kindly ask that you read the paper again because there's a lot of misinterpretations about the paper in the review.
>
> # Introduction
> It's not true that policy gradient based methods suffer more from the model bias. Model based is a family of methods. Some specific methods/implementations may suffer more from the model bias than the others. However, generalizing that Q-value function methods are more robust to the ones that are built on top of policy gradient is simply inaccurate and unfounded.
>
> It’s certainly possible that we can extract a parametric policy from the optimization that MPC performs but this is an indirect process. Indirect training may lead to loss of accuracy. Furthermore, training a parametric policy from an MPC policy can only be done after having a final MPC policy and hence doesn't alleviate the problem of having to do solve the hard optimization problem during the decision time.
>
> Your claim that “The MPC will transfer better to other tasks that have the same dynamics, since it is not task specific ...” is not true and mischaracterizes our method. There are two ways to do task transfers:
> One is the dynamics model. All model-based methods, including ours, support this concept. Dynamics model transfer has nothing to do with MPC.
> Another is the policy transfer. Two similar tasks may induce two similar policies. Hence, it’s worthwhile being able to transfer the policy. MPC is an implicit policy, by way of solving an optimization problem each time the agent has to make a decision, and therefore harder to do policy transfer.
>
> # Related work
> We thought that we had covered all relevant related work. It’s true that “risk-sensitive” and “optimization in the face of uncertainty” are not novel terminologies. Those are just ambiguous labels. According to our best knowledge, the way that we fully quantify the uncertainty of the value function estimate, via the uncertainty of the model, and the way that we combine return and risk in a unified objective function is unique in the literature of RL.
>
> We would appreciate if you can highlight some relevant related work that we should add to our references.
>
> # Uncertainty-Aware Model-Based Policy Optimization
> While each component (risk-sensitive objective, bootstraps, rollout) of POUM is well studied, the primary contribution of this work is to combine rollouts techniques and bootstrap models to fully propagate uncertainty into value function V(\pi)(s). Then V(\pi)(s) is transformed into a deterministic utility function U(\pi)(s) that reflects a subjective measure balancing the risk and return.
>
> Note that policy gradient is not a method. In the context of our paper, policy gradient denotes a class of policy optimization methods that rely on computing the gradient of the V as a function of \pi.
>
> Our method of computing the policy gradient is entirely different from all prior policy gradient works (REINFORCE, TRPO, PPO, etc.). To our knowledge, no method has tried to compute the policy gradient by backprop through the chained models (dynamics model, policy model, and reward model).
>
> Using a deterministic policy is not a key idea of the paper and we didn’t claim that it's a novelty. In section 3.3.2: we only argued for this choice. While all estimations, including that of the dynamics model and of the  value function, need to be stochastic, i.e. uncertainty aware, the policy does not need to be. Deterministic policy simply means that the agent is consistent when taking an action, no matter how uncertain it may be about the world.
>
> ===========
> #Experiment
> ===========
> We compare our algorithm to two model-based methods: MBPO and STEVE. There are also other works on MBRL including ME-TRPO [6],  PETS [7], MB-MPO [8]..., but their public implementations couldn’t run on standard OpenAI gym environments that we rely on. For instance, several of them relied on the modified MuJoCo environments in RLLab, which makes it easier to train an agent on Half Cheetah.
>
> Hence, we couldn’t do a fair comparison against those methods.
>
> As an aside, this paper [https://arxiv.org/abs/1907.02057v1] is another example of why benchmarking many model-based RL methods is not easy and has been published as a standalone paper. Even in that work, the comparisons are not entirely fair. This GitHub issue shows an example: [https://github.com/WilsonWangTHU/mbbl/issues/2]
>
>
> [7] Deep reinforcement learning in a handful of trials using probabilistic dynamics models. Kurtland Chua, Roberto Calandra, Rowan McAllister, and Sergey Levine.
> [8] Model-based reinforcement learning via meta-policy optimization. Ignasi Clavera, Jonas Rothfuss, John Schulman, Yasuhiro Fujita, Tamim Asfour, and Pieter Abbeel.

---

### Decision · Program_Chairs · 2019-12-19

**Decision:**

Reject

**Comment:**

The main contribution of this work is introducing the uncertainty-aware value function prediction into model-based RL, which can be used to balance the risk and return empirically.

The reviewers generally agree that this paper addresses an interesting problem, but there are some concerns that remain (see reviewer comments).

I also want to highlight that in terms of empirical results, it is insufficient to present results for 3 different random seeds. To highlight any kind of robustness, I suggest *at least* 10-20 different random seeds; otherwise the findings can/will be misleading.